

# The prognostic and clinicopathological significance of desmoglein 2 in human cancers: a systematic review and meta-analysis

Jiantao Wang[1,2,*], Siyuan Hao[3,*], Junjie Gu[3], Sean G. Rudd[2] and Yan Wang[3]

[1] State Key Laboratory of Biotherapy & Department of Lung Cancer Center and Department of Radiation Oncology, West China Hospital, Sichuan University, Chengdu, China
[2] Science for Life Laboratory, Department of Oncology-Pathology, Karolinska Institutet, Stockholm, Sweden
[3] State Key Laboratory of Oral Diseases, National Clinical Research Center for Oral Diseases, West China Hospital of Stomatology, Sichuan University, Chengdu, China
* These authors contributed equally to this work.

## ABSTRACT

**Objective:** The survival and clinicopathological significance of desmoglein 2 (DSG2) in various cancers is controversial. Thus, we performed this systematic review and meta-analysis to explore the preliminary prognostic value of DSG2.

**Methods:** Eligible studies were identified from databases including PubMed, the Cochrane Library, Embase, Web of Science and Scopus. Hand searches were also conducted in relevant bibliographies. We then extracted and pooled hazard ratio (HR) of overall survival (OS) and odds ratio (OR) of clinicopathological features.

**Results:** A total of 11 eligible studies containing 1,488 patients were included. Our results demonstrated that in non-small cell lung cancer (NSCLC), high DSG2 expression is associated with poor OS. However, in digestive system cancer and female reproductive system cancer, there were no statistically significant associations between OS and DSG2.

**Conclusions:** Based on the findings of this study, high DSG2 expression is associated with worse prognosis in patients with NSCLC, and thus DSG2 expression could be a biomarker for prognosis in NSCLC.

## INTRODUCTION

Cancer is the leading cause of death worldwide, and with increasing incidence, is becoming a severe public heath burden. Despite many advances in our understanding of this disease, it is still uncommon for cancer patients to be diagnosed at an early stage, and instead, the majority of patients are initially diagnosed at an advanced stage. Additionally, the prognosis of these cancer patients remains poor due to limited efficacy of conventional treatments such as chemoradiotherapy. Therefore, it is of great value to identify factors closely related to cancer initiation and progression that can be used as effective biomarkers

Corresponding author
Yan Wang,
wangyan1458@scu.edu.cn

for early diagnosis, prognosis evaluation, and treatment guidance in cancer patients (*Siegel et al., 2021*).

Desmoglein 2 (DSG2), a trans-membrane glycoprotein belonging to the cadherin superfamily, mediates cell-cell adhesion *via* intact desmocollin to form desmosome sand and plays an important role in maintaining epithelial tissue integrity. Recently, several studies reported that aberrant expression of DSG2 was associated with tumorigenesis and cancer progression in a variety of malignancies. However, the exact role of DSG2 in cancer remains unclear and appears to be complicated, with data indicating it can function as both a tumor suppressor and an oncogene in different cancer types. For example, *Chang et al. (2021)* reported that high expression of DSG2 increased cancer cell invasiveness and led to worse prognosis in breast cancer, whilst the opposite was observed in another study, which found that knockdown of DSG2 in anaplastic thyroid cancer cells increased cell migration and distant metastasis in xenografts (*Lee et al., 2020*). On the issue of relation between prognosis and DSG2 expression, existing studies also showed different results. In NSCLC, *Cai et al. (2017)* and *Jin et al. (2020)* reported that high expression of DSG2 was associated with unfavorable prognosis and a negative correlation between DSG2 expression and prognosis was also found by *Han et al. (2018)* in hepatocellular carcinoma. However, in esophageal squamous cell carcinoma, DSG2 expression showed no significant association with prognosis (*Fang et al., 2014*) and in ovarian serous carcinoma, *Chen et al. (2018)* reached the reverse conclusion that low expression of DSG2 was related to poor prognosis. Taken together, the relationship between DSG2 expression and prognosis in cancer patients is inconclusive and is yet to be systematically reviewed. Thus, this study aimed to define the prognostic value of DSG2 in patients with cancer.

## METHODS

The study protocol was registered on the PROSPERO database (http://www.crd.york.ac.uk/PROSPERO) with ID: CRD42020202201. The Preferred Reporting Items for Systematic Reviews and Meta-Analyses (PRISMA) was followed (*Moher et al., 2009*).

### Search strategy

Two authors (Jiantao Wang and Siyuan Hao) searched a number of electronic databases, including PubMed, the Cochrane Library, Embase, Web of Science and Scopus, to identify relevant studies using the following key words: Desmoglein 2, Neoplasms, Tumor, Cancer and Malignancy. References of the studies were also checked to identify potentially relevant papers. Our detailed search strategy is exhibited in Table 1.

### Inclusion and exclusion criteria

Studies included in this systematic review and meta-analysis are required to meet the following criteria: (1) The relationship between the expression of DSG2 and cancer was investigated; (2) Patients were diagnosed with cancer *via* pathology, including biopsy, HE staining, professional pathologist diagnosis, and genetic testing if necessary;

**Table 1 Our search strategy of PubMed, the Cochrane Library, Embase, Web of Science and Scopus.** The search time limit is from the establishment of databases to January 31st 2021.

| Electronic database and search strategy | |
|---|---|
| PubMed | #1 ((((desmoglein 2[MeSH Terms]) OR (Desmoglein II[Title/Abstract])) OR (Desmoglein-2[Title/Abstract])) OR (Desmosomal Glycoprotein 2[Title/Abstract]) <br> #2 ((((((((((((((((Neoplasms[MeSH Terms]) OR (Neoplasia[Title/Abstract])) OR (Neoplasias[Title/Abstract])) OR (Neoplasm [Title/Abstract])) OR (Tumors[Title/Abstract])) OR (Tumor[Title/Abstract])) OR (Cancer[Title/Abstract])) OR (Cancers[Title/Abstract])) OR (Malignancy[Title/Abstract])) OR (Malignancies[Title/Abstract])) OR (Malignant Neoplasms[Title/Abstract])) OR (Malignant Neoplasm[Title/Abstract])) OR (Neoplasm, Malignant[Title/Abstract])) OR (Neoplasms, Malignant[Title/Abstract])) OR (Benign Neoplasms[Title/Abstract])) OR (Neoplasms, Benign[Title/Abstract])) OR (Benign Neoplasm[Title/Abstract])) OR (Neoplasm, Benign[Title/Abstract]) <br> #3 #1 AND #2 |
| Cochrane Library | #1 MeSH descriptor: [Desmoglein 2] explode all trees <br> #2 MeSH descriptor: [Neoplasm] explode all trees <br> #3 #1 AND #2 |
| Embase | #1 'malignant neoplasm'/exp <br> #2 'tumor':ab,ti <br> #3 'neoplasm':ab,ti <br> #4 'carcinoma':ab,ti <br> #5 'cancer':ab,ti <br> #6 'desmoglein 2'/exp <br> #7 'desmoglein 2':ab,ti <br> #8 #1 OR #2 OR #3 OR #4 OR #5 <br> #9 #6 OR #7 <br> #10 #8 AND #9 |
| Web of Science | AB = (Desmoglein 2 OR Desmoglein II OR Desmosomal Glycoprotein 2) AND AB = (Neoplasms OR Tumor OR Cancer OR Malignant Neoplasm) <br> Indexes = SCI-EXPANDED, SSCI, A&HCI, CPCI-S, CPCI-SSH, ESCI, CCR-EXPANDED, IC Timespan = 1975 to January 31st 2021 |
| Scopus | (TITLE-ABS-KEY ('malignant neoplasm') OR TITLE-ABS-KEY ('tumor') OR TITLE-ABS-KEY ('neoplasm') OR TITLE-ABS-KEY ('carcinoma') OR TITLE-ABS-KEY ('cancer')) AND (TITLE-ABS-KEY ('desmoglein 2')) |

(3) The expression of DSG2 was detected by immunohistochemistry (IHC) or real-time quantitative polymerase chain reaction (RT-qPCR); (4) The hazard ratios (HR) with 95% confidence intervals (CI) can be calculated directly using reported data or can be extracted from the survival curve provided. Studies will be excluded on account of any of the following reasons: (1) The follow-up and survival data were not reported and cannot be retrieved by contacting the author; (2) Sample size of the study was less than 40; (3) Review, meta-analysis, case-report and *in vitro* studies; (3) Studies published in non-English language.

## Quality assessment of included studies

The Newcastle-Ottawa Scale (NOS) was used to assess the quality of the included studies. Ranging from one to nine, a score of six and above was regarded as high quality. Two investigators (Jiantao Wang and Siyuan Hao) independently finished the assessment. Disagreement was first settled through discussion. If there was any disagreement, it would be solved by the discussion between Jiantao Wang and Siyuan Hao. Unresolved divergence was discussed with a third investigator (Yan Wang) and decided by Yan Wang.

## Data extraction

The following information was extracted from the scanned articles: author's name, date published, cancer type, number of cases, age, follow-up time, detection method, cut-off value, percentage of high expression patients and reported outcomes. We also paid attention to clinicopathological features, including age, sex, differentiation status, lymph node metastasis and TNM stage. For clinicopathological features, we calculated the OR and their 95% CI from the data provided or obtained from the author. For survival data, HR and 95% confidence interval (CI) of DSG2 expression for OS were extracted. If the 95% CI was not provided, we would calculate it through a published method which was published by *Tierney et al. (2007)*.

## Statistical analysis

The relationship between the expression of DSG2 and the prognosis of cancer was evaluated by the pooled HR and their 95% confidence interval (CI) for OS. If the pooled HR > 1 and the lower limit of 95% CI > 1, it suggested that poor prognosis is more likely to occur in patients with high DSG2 expression. Pooled OR and 95% CI were calculated to estimate the association between clinicopathological features and the expression of DSG2. In the meta-analysis, Cochrane's Q statistics and the Higgins I-squared statistic ($I^2$) were used to detect heterogeneity. $P < 0.1$ or $I^2 > 50\%$ was regarded as statistically significant heterogeneity, and we applied a random effect model. A fixed effect model was preferable when $P > 0.05$ or $I^2 < 50\%$. For significant heterogeneity, we carried out subgroup analysis and sensitivity analysis to resolve it. All these analyses were performed with RevMan 5.3 software.

# RESULTS

## Search results

We identified 116 studies from PubMed, 0 from the Cochrane Library, 212 from Embase, 181 from Web of Science and 151 from Scopus database. Twelve studies were also found through the references. We had a total of 358 articles after removing duplication. First, after reading the title and abstract, 279 articles were excluded for irrelevance. Then, we read the full text of the remaining 79 articles, among which 41 *in vivo* studies, 21 conference abstracts, four reviews and two bioinformatics analyses were excluded. Ultimately, 11 studies (*Barber et al., 2014*; *Cai et al., 2017*; *Chen et al., 2018*; *Fang et al., 2014*; *Han et al., 2018*; *Jin et al., 2020*; *Ormanns et al., 2015*; *Qin et al., 2020*; *Sun et al., 2020*; *Xu et al., 2020*; *Yashiro, Nishioka & Hirakawa, 2006*) containing 1,488 cases were selected for systematic review, and 10 studies were included in the meta-analysis (Fig. 1).

## Study characteristics

The main characteristics extracted from the studies, including the author's name, date published, cancer type, number of cases, age, follow-up time, detection method, cut-off value, percentage of high expression patients and reported outcomes, are summarized in Table 2. There were 11 types of cancer, including prostate cancer, NSCLC, lung
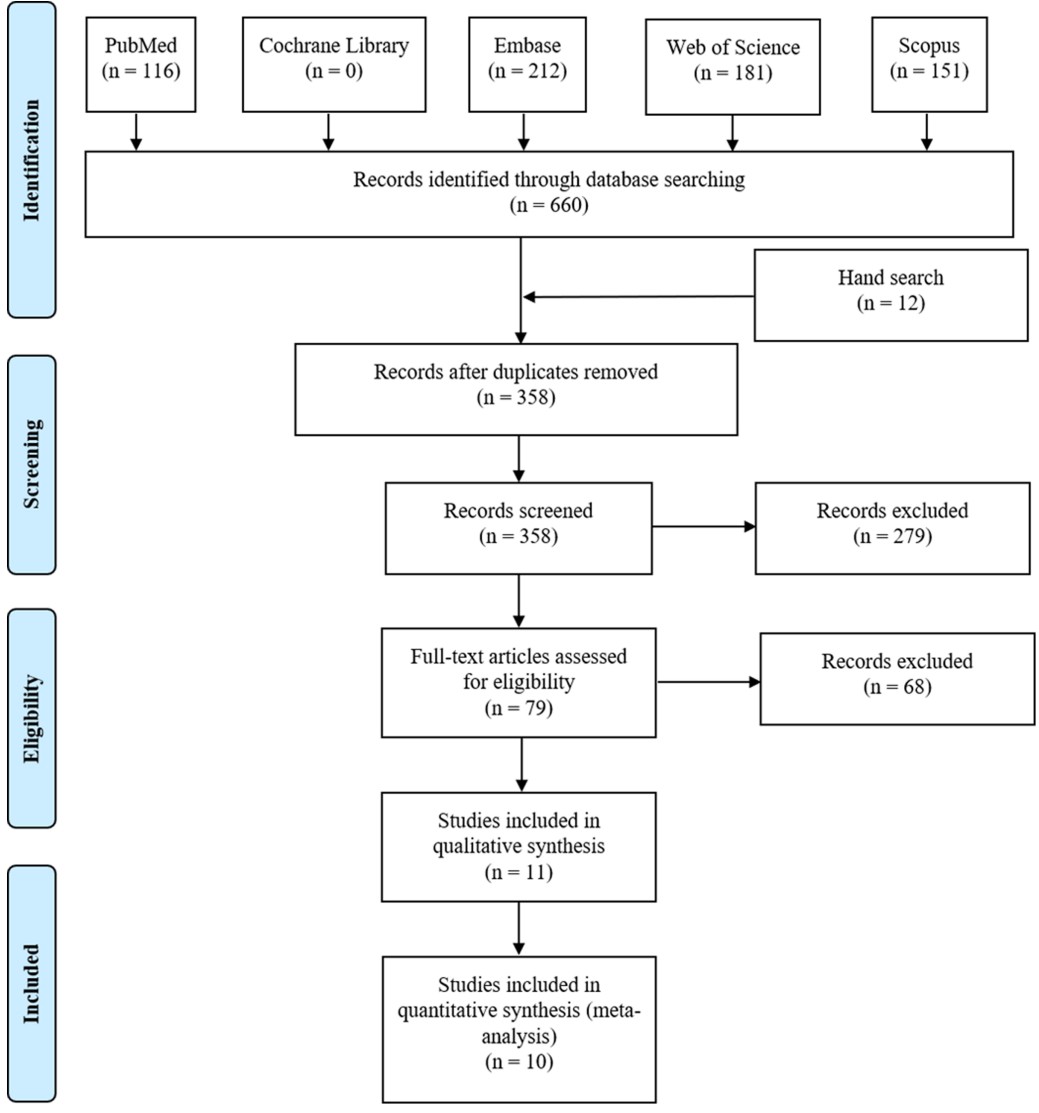

**Figure 1  The flow plot of our literature search.** We searched electric databases including PubMed, The Cochrane Library, Embase, Web of Science and Scopus. Hand searches were also conducted in relevant bibliographies. PRISMA was followed to identify eligible studies to include.

adenocarcinoma, high-grade ovarian serous carcinoma, esophageal squamous cell carcinoma, lung adenocarcinoma, pancreatic ductal adenocarcinoma, cervical cancer, extrahepatic cholangiocarcinoma, hepatocellular carcinoma and diffuse-type gastric cancer. Each of the included studies contains more than 40 patients. The methods to detect DSG2 expression were immunohistochemistry (IHC) and real-time quantitative polymerase chain reaction (RT-qPCR). The results of survival data for the included studies were obtained through follow-up and reported in the form of OS, progression-free survival (PFS) or recurrence-free survival (RFS).
**Table 2 Characteristics of the included studies.**

| Study | Cancer type | Expression | Detecting method | Sample size | Mean age (range) | Follow-up (month) | Cut-off | High expression (%) | Endpoints |
|---|---|---|---|---|---|---|---|---|---|
| Barber et al. (2014) | Prostate cancer | protein | IHC | 414 | 61 (41–74.7) | 123.6 | 60% | 40 | RFS |
| Cai et al. (2017) | Non-small cell lung cancer | protein | IHC | 70 | 61 (39–77) | 94 | score ≥3 | 66 | OS |
| Chen et al. (2018) | High-grade ovarian serous carcinoma | protein | IHC | 162 | 54.9 (18–72) | 91 | score ≥5 | 38 | OS, PFS |
| Fang et al. (2014) | Esophageal squamous cell carcinoma | mRNA | RT-qPCR | 85 | 55 (40–88) | 60 | tumor/normal >0.5 fold | 76 | OS |
| Han et al. (2018) | Hepatocellular carcinoma | mRNA | RT-qPCR | 104 | 50 | 60 | tumor/normal >1.0 fold | 65 | OS |
| Jin et al. (2020) | Lung adenocarcinoma | protein | IHC | 86 | 60 (43–78) | 60 | score ≥2 | 75 | OS |
| Ormanns et al. (2015) | Pancreatic ductal adenocarcinoma | protein | IHC | 165 | 66 (33–87) | 60 | score ≥2 | 65 | OS |
| Qin et al. (2020) | Cervical cancer | protein | IHC | 150 | 42 | 60 | score ≥6 | 40 | OS |
| Xu et al. (2020) | Extrahepatic cholangiocarcinoma | protein | IHC | 100 | 58.8 (35–80) | 30 | stained cells ≥25% | 47 | OS |
| Yashiro, Nishioka & Hirakawa (2006) | Diffuse-type gastric cancer | protein | IHC | 112 | 61.4 | 107 | stained cells >20% | 71 | OS |
| Sun et al. (2020) | Lung adenocarcinoma | mRNA | RT-qPCR | 40 | NA | 100 | score ≥7 | 50 | OS, PFS |

## Quality assessment

Each study's score of the NOS is shown in Table 3. According to the NOS, the quality of the majority of included studies was relatively high, with a mean value of 7.3. Among them, two achieved a score of 6 (Chen et al., 2018; Fang et al., 2014), four achieved a score of 7 (Cai et al., 2017; Ormanns et al., 2015; Sun et al., 2020; Xu et al., 2020) and five achieved a score of 8 (Barber et al., 2014; Han et al., 2018; Jin et al., 2020; Ormanns et al., 2015; Yashiro, Nishioka & Hirakawa, 2006). A score not less than 6 is regarded as high quality. Generally, the included studies were recognized to have a low risk of bias in cohort studies.

## Association between DSG2 expression and OS

In total, 10 studies containing 1,074 patients were included in the meta-analysis to assess the association between DSG2 expression and OS (Cai et al., 2017; Chen et al., 2018; Fang et al., 2014; Han et al., 2018; Jin et al., 2020; Ormanns et al., 2015; Qin et al., 2020; Sun et al., 2020; Xu et al., 2020; Yashiro, Nishioka & Hirakawa, 2006). The pooled HR suggested no statistical significance (HR = 1.02, 95% CI [0.75–1.40], $I^2$ = 88%, random-effect model) (Fig. 2). Considering that the cancer types were different, we further conducted subgroup analysis to explore the association between DSG2 expression and OS in different cancer types. Studies were divided into three subgroups based on cancer types: NSCLC (Cai et al., 2017; Jin et al., 2020; Sun et al., 2020), digestive system cancer
**Table 3 Result of quality assessment of included studies according to Newcastle-Ottawa Scale (NOS) for cohort study.**

| Study | Selection | | | | Comparability | | Outcome | | | Total score |
|---|---|---|---|---|---|---|---|---|---|---|
| | S1 | S2 | S3 | S4 | C1 | C2 | O1 | O2 | O3 | |
| Barber et al. (2014) | + | + | + | + | + | + | + | | + | 8 |
| Cai et al. (2017) | + | + | | + | + | + | + | | + | 7 |
| Chen et al. (2018) | + | + | | + | + | + | + | | | 6 |
| Fang et al. (2014) | + | + | | + | + | + | + | | | 6 |
| Han et al. (2018) | + | + | + | + | + | + | + | | + | 8 |
| Jin et al. (2020) | + | + | + | + | + | + | + | | + | 8 |
| Ormanns et al. (2015) | + | + | + | + | + | + | + | | + | 8 |
| Qin et al. (2020) | + | + | | + | + | + | + | | + | 7 |
| Xu et al. (2020) | + | + | | + | + | + | + | | + | 7 |
| Yashiro, Nishioka & Hirakawa (2006) | + | + | + | + | + | + | + | | + | 8 |
| Sun et al. (2020) | + | + | | + | + | + | + | | + | 7 |

**Note:**
+ Means that the condition required by Newcastle-Ottawa Scale (NOS) for cohort study is met and is recorded as one point. S1: Representativeness of the exposed cohort, S2: Selection of the non-exposed cohort, S3: Ascertainment of exposure, S4: Demonstration that outcome of interest was not present at start of stud, C1: According to the most important factor to choose control, C2: According to the other important factor to choose control, O1: Assessment of outcome, O2: Follow-up long enough for outcomes to occur, O3: Adequacy of follow-up of cohorts.

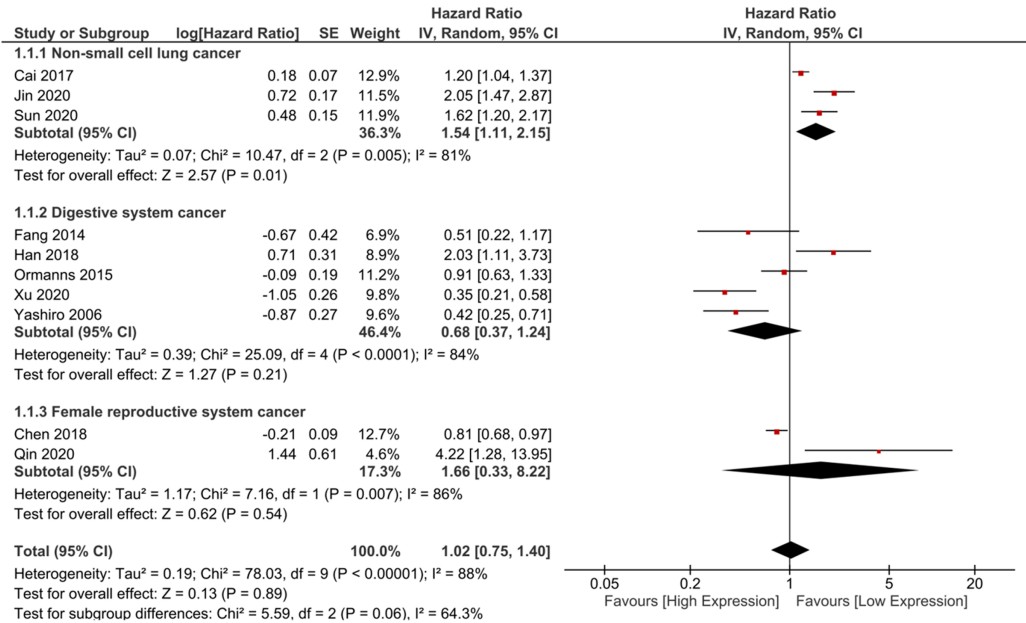

**Figure 2 The forest plot of meta-analysis.** We combined the results of included studies and the total result was not statistically significant. Subgroup analysis indicated that high expression of DSG2 was significantly associated with poor prognosis, but not significant in digestive system cancer and female reproductive system cancer.

(*Fang et al., 2014*; *Han et al., 2018*; *Ormanns et al., 2015*; *Xu et al., 2020*; *Yashiro, Nishioka & Hirakawa, 2006*) and female reproductive system cancer (*Chen et al., 2018*; *Qin et al., 2020*). Barber's study was not included in the meta-analysis because it only reported RFS (*Barber et al., 2014*). The results of the meta-analysis showed that high DSG2

**Table 4** The association between DSG2 and clinicopathological features in NSCLC and digestive system cancer.

| Clinicopathologic parameters | Studies | Cases | Quantitative synthesis | | | Test for heterogeneity | | | |
|---|---|---|---|---|---|---|---|---|---|
| | | | Pooled OR | 95% CI | P value | $\chi^2$ | P value | I$^2$ (%) | Model |
| **NSCLC** | | | | | | | | | |
| Age (≥60 vs. <60) | 3 | 196 | 0.97 | [0.56–1.69] | 0.92 | 0.94 | 0.63 | 0 | Fixed |
| Gender (male vs. female) | 3 | 196 | 2.33 | [0.84–6.42] | 0.10 | 5.46 | 0.07 | 63 | Random |
| Tumor size (≥3 cm vs. <3 cm) | 2 | 156 | 0.59 | [0.16–2.24] | 0.44 | 2.60 | 0.11 | 62 | Random |
| Differentiation (poor vs. well/moderate) | 2 | 156 | 1.43 | [0.96–2.96] | 0.33 | 1.41 | 0.24 | 29 | Fixed |
| Lymph node metastasis (yes vs. no) | 2 | 156 | 0.95 | [0.49–1.84] | 0.88 | 0.14 | 0.71 | 0 | Fixed |
| TNM stage (III/IV vs. I/II) | 3 | 196 | 1.04 | [0.79–2.05] | 0.25 | 2.29 | 0.32 | 12 | Fixed |
| **Digestive system cancer** | | | | | | | | | |
| Tumor size (≥3 cm vs. <3 cm) | 3 | 289 | 1.19 | [0.27–1.96] | 0.49 | 1.35 | 0.51 | 0 | Fixed |
| Differentiation (poor vs. well/moderate) | 3 | 297 | 0.27 | [0.10–0.78] | 0.02 | 4.12 | 0.13 | 51 | Random |
| Lymph node metastasis (yes vs. no) | 4 | 462 | 0.47 | [0.22–0.99] | 0.05 | 8.87 | 0.03 | 66 | Random |
| TNM stage (III/IV vs. I/II) | 3 | 289 | 0.51 | [0.18–1.49] | 0.22 | 8.06 | 0.02 | 75 | Random |

expression was significantly associated with poor OS in NSCLC (HR = 1.54, 95% CI [1.11–2.15], I$^2$ = 81%, random-effect model). In digestive system cancer, DSG2 expression was not statistically correlated with OS (HR = 0.68, 95% CI [0.37–1.24], I$^2$ = 84%, random-effect model). In female reproductive system cancer, the result also indicated no statistical significance (HR = 1.66, 95% CI [0.33–8.22], I$^2$ = 86%, random-effect model).

## Association between DSG2 expression and clinicopathological features NSCLC

The association between DSG2 and clinicopathological features in NSCLC is shown in Table 4. We pooled the OR of clinicopathological parameters, including age (≥60 vs. <60), sex (male vs. female), tumor size (≥3 cm vs. <3 cm), differentiation status (poor vs. well/moderate), lymph node metastasis (yes vs. no) and TNM stage (III/IV vs. I/II). However, the results of the meta-analysis indicated no statistical significance.

## Digestive system cancers

The association between DSG2 and clinicopathological features in digestive system cancer is also shown in Table 4. We pooled the OR of clinicopathological parameters, including tumor size (≥3 cm vs. <3 cm), differentiation status (poor vs. well/moderate), lymph node metastasis (yes vs. no) and TNM stage (III/IV vs. I/II). The results demonstrated that DSG2 was correlated with lymph node metastasis and TNM stage in digestive system cancer.

## Publication bias and sensitivity analysis

To detect the publication bias among included studies, we mapped the funnel plot in RevMan software (Fig. 3). In the funnel plot, the points on both sides were evenly distributed, indicating on significant publication bias among the included studies. We also did the sensitivity analysis to detect the robustness of our result. When we omitted the

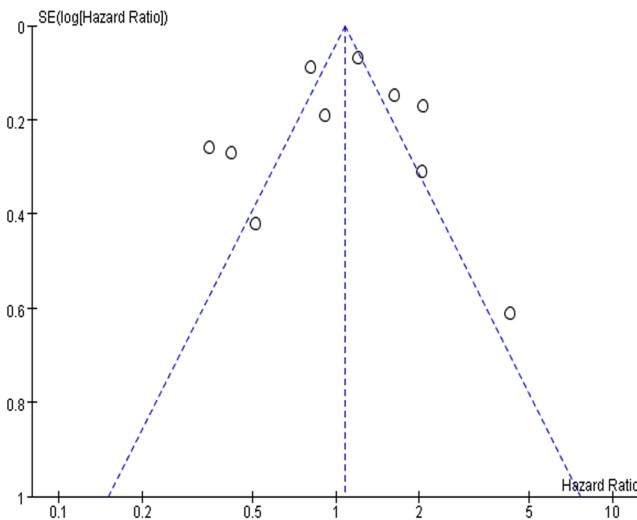

**Figure 3 The funnel plot of publication bias.** The points in the plot represents the result of each included studies. They are symmetrical distributed on both side, indicating no significant publication bias.

included studies one by one in RevMan software, the result of meta-analysis did not dramatically change, showing that our result was stable and robust (Supplemental File 1).

## DISCUSSION

Cell-cell anchoring junctions are important for tumor cell biology and have been gaining increased attention (*Wu et al., 2019*). It has been reported that cell-cell interactions *via* E-cadherin can protect cancer cells from ferroptosis, an iron-dependent programmed cell death pathway, whilst integrin-mediated anchoring can lead to metabolic reprogramming to support cancer cell survival (*Endo et al., 2020*). DSG2, a key factor that mediates cell-cell junctions, was also reported to regulate cancer cell behavior and could be a biomarker for prognosis in cancer patients. In this study, we performed a systematic review and meta-analysis that included 11 studies with 1,488 patients in total. The combined data from all 10 included studies showed no correlation between DSG2 expression and prognosis. Considering the heterogeneity in the 10 included studies for the meta-analysis, which might lead to unreliable conclusions, we conducted further subgroup analysis. We assessed the association between DSG2 expression and the prognosis of cancers in NSCLC, digestive system cancer and female reproductive system cancer. In NSCLC, high DSG2 expression was significantly associated with unfavorable survival. Together with the pooled OS and OR of the indicated clinicopathological features, our analyses suggested DSG2 may function as a promoter of tumor progression, which is supported by our mining of the TCGA database. Furthermore, the TCGA database mining also indicated that high DSG2 expression was associated poor prognosis in NSCLC (Supplemental File 2). In digestive system cancers, there was no statistically significant association between DSG2 and survival. There was only one study in male reproductive system cancer and two studies in female reproductive system cancers, thus, due to

insufficient evidence; we have not yet been able to draw convincing conclusions in these two subgroups.

The function of DSG2 is not limited to the cell junction (*Chidgey & Dawson, 2007*). According to recent findings, the mechanisms of DSG2 in regulating cancer development and the response to anticancer treatment are associated with cell cycle changes and molecular interactions involved in signal transduction (*Hao, Liu & Ma, 2021*; *Ungewiß et al., 2018*). In NSCLC, *Cai et al. (2017)* reported that DSG2 knockdown inhibited NSCLC xenograft growth in nude mice. Knockdown of DSG2 reduced NSCLC cell proliferation and arrested them at the G1 phase of the cell cycle. In addition, high DSG2 expression was also reported to promote cancer cell growth and migration, and decrease the response sensitivity to EGFR targeting therapy in NSCLC cancer cells. A further study revealed that it was mediated *via* DSG2-activated Src and the downstream Rac1-PAK1 signaling pathway by binding and stabilizing EGFR in the cell membrane in NSCLC cancer cells (*Jin et al., 2020*). Therefore, we consider that its effect on the cell cycle and interaction with EGFR contribute to the unfavorable prognosis of NSCLC. Unlike lung cancer, the role of DSG2 in digestive cancers was not well defined and previous reports were even paradoxical. For instance, *Han et al. (2018)* reported that DSG2 expression was significantly higher in hepatocellular carcinoma tumor tissues than in matched noncancerous tissues and positively correlated with tumor size, tumor stage and unfavorable prognosis, while *Xu et al. (2020)* illustrated that the expression of DSG2 proteins was significantly lower in extrahepatic cholangiocarcinoma tissues than in normal tissues and that negative DSG2 expression was an independent poor prognosis factor. In anaplastic thyroid cancers, Lee demonstrated that DSG2 knockdown induced cell motility and invasiveness through the cMet/Src/Rac1 signaling axis and that targeting cMet inhibited DSG2 knockdown-induced distant metastasis in mouse xenografts (*Lee et al., 2020*). All these studies indicated a differing role for DSG2 in cancer development, progression and response to anticancer therapy, and thus further studies are needed to uncover the mechanisms at play.

Admittedly, there are limitations to our systematic review and meta-analysis. First, the number of included studies was relatively limited. Second, the sample size of each included study was also limited, and all eligible studies were retrospective studies. Third, it was difficult to determine a standard expression cut-off value because of different cancer types and diverse detection methods in the included studies, which may also lead to heterogeneity among the included studies. Therefore, further high-quality clinical studies are required to attain a stable conclusion.

## CONCLUSION

The association between DSG2 expression and prognosis was not consistent between different types of cancers. Our study demonstrated that in NSCLC, high DSG2 expression was associated with poor prognosis. Therefore, DSG2 can potentially be a biomarker of the prognosis of NSCLC.

### Funding

This study was supported by the Scientific Research Foundation of the Health Planning Committee of Sichuan (18PJ186 to Jiantao Wang), the National Natural Science Foundation of China (81600864 to Yan Wang), the Undergraduate Student Innovation and Entrepreneurship Training Program of Sichuan University (C202114146 to Siyuan Hao), and the Swedish Cancer Society (19-0056-JiA to Sean G. Rudd). The funders had no role in study design, data collection and analysis, decision to publish, or preparation of the manuscript.

### Grant Disclosures

The following grant information was disclosed by the authors:
Scientific Research Foundation of the Health Planning Committee of Sichuan: 18PJ186.
National Natural Science Foundation of China: 81600864.
Undergraduate Student Innovation and Entrepreneurship Training Program of Sichuan University: C202114146.
Swedish Cancer Society: 19-0056-JiA.

### Competing Interests

The authors declare that they have no competing interests.

### Author Contributions

- Jiantao Wang conceived and designed the experiments, performed the experiments, analyzed the data, authored or reviewed drafts of the paper, and approved the final draft.
- Siyuan Hao performed the experiments, analyzed the data, prepared figures and/or tables, and approved the final draft.
- Junjie Gu analyzed the data, prepared figures and/or tables, and approved the final draft.
- Sean G. Rudd conceived and designed the experiments, authored or reviewed drafts of the paper, and approved the final draft.
- Yan Wang conceived and designed the experiments, authored or reviewed drafts of the paper, and approved the final draft.

### Data Availability

The raw measurements are available in the Supplemental Files.

### Supplemental Information

Supplemental information for this article can be found online at http://dx.doi.org/10.7717/peerj.13141#supplemental-information.

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
