# Peer review of "The prognostic and clinicopathological significance of desmoglein 2 in human cancers: a systematic review and meta-analysis"

_PeerJ, doi:10.7717/peerj.13141_

## Round 0.1 · original submission · Major Revisions

Please address the concerns of both reviewers and revise your manuscript accordingly.

Reviewer 1 ·

Basic reporting

In the article the authors have reviewed the current literature to establish the prognostic value of DSG2. The results are interesting and is useful to understand the relationship between the level of DSG2 expression and prognosis of cancer patients.

The article is written in a simple way which is easy for the readers to understand. Spaces are missing at some sentences (such as line 31, between 'in' and 'the'; line 34, in between 'to' and 'cancer'; line 42, in between 'of' and 'DSG2'; and few other places). Please go through the manuscript once thoroughly for such minor errors.

The discussion part has enough literature survey but I would strongly recommend including a little more detail about the articles included in the review in the introduction section.

The figures and tables are useful for understanding the result. However, the table headlines are a little messy. If possible please change the font size and column width so that the words fit properly in the same line.

The review, especially the subset studies seems interesting and can lead to further useful research on this topic.

Experimental design

The selection of the articles and the filtering process has been explained in a detail and easy-to-understand way. The only thing I would suggest is adding a little detail about the 'invalid line' mentioned in line 91.

Validity of the findings

The validity of findings are interesting. Additionally, this finding can lead to clinical studies on the topic.
The discussion section is very well written with useful citations and valid points on the limitations of the current work.

Additional comments

I would strongly recommend adding a list of abbreviations used for readers' convenience.
In line 118 what do you mean by 'The included studies contained more than 40 cases'?
The reference format of Chen L, 2018 (line 243) seems a little different to me. Please cross check once.

With the above mentioned minor revisions taken care of, the manuscript has enough contents to be published in PeerJ.

Reviewer 2 ·

Basic reporting

Wang and colleagues have investigated the prognostic potential of DSG2 in different types of cancer by a systematic review and meta-analysis. Although this is an interesting topic, there are several major issues that should be taken into consideration before the paper is published.
There are several grammatical and syntax errors, and typos (e.g., “no-small” (line 149), “meat-analysis” (line 294)). The entire manuscript should undergo thorough revision. The figure captions should also be more self-explanatory; more detailed description is needed.

Experimental design

Regarding Methodology, a more adequate description is required. For example, in the keywords it is not clear the alternative wording and the combination of different words used; also, some major informative keywords are missing like “survival” or “prognosis” or “survival”. It is not clear what the authors mean by “pathological methods”, and also whether any studies published in a non-English language were included in the analysis. The authors do not explain how they extracted data from the survival curves.

Validity of the findings

The major limitation of this study is the rather limited number of eligible studies. Therefore, any inferences or conclusions made should not be hasty. The authors could also supplement their study with relevant data/information from other sources like The Cancer Genome Atlas (TCGA).
In order to improve the accuracy of their results, the authors could have performed Sensitivity analysis of each eligible study or assessed publication bias.

---

## Round 0.2 · Minor Revisions

Please address remaining issues pointed by reviewer #2 and amend your manuscript accordingly.

Reviewer 1 ·

Basic reporting

The manuscript has been improved significantly.

Experimental design

Improved.

Validity of the findings

Improved.

Additional comments

All the suggestions have been included satisfactorily. I would recommend accepting the publication.

Reviewer 2 ·

Basic reporting

The comments have been taken into consideration and addressed adequately by the authors. Yet, English language should be improved. For example, Line 50: ...as it can be functional...” could be changed to “can function as both...”; Line 80: diagnosed with cancer via pathology...” A more accurate technical term should be used to describe the method used for cancer diagnosis. Moreover, assertive language should be avoided like “Based on currently available evidence, the association of DSG2 expression with prognosis was context dependent.” Given also the relatively small number of eligible studies, this sentence could be revised “Based on the findings of this study, high DSG2 expression is associated with worse prognosis in patients with nonsmall-cell lung cancer.”

Experimental design

The methods are described quite sufficiently. However, more explanation is needed regarding the overall survival data derived from TCGA, that is, the clinical features of the selected patients (days to death?) , as well as the R package used for data processing and analysis (join.Cox package?)

Validity of the findings

All data used in the meta-analysis are provided and they are statistically sound.
The Conclusions also largely support the findings of this study.

---

## Round 0.3 · accepted · Accept

Remaining issues pointed by the reviewer were adequately addressed and revised manuscript is acceptable now.